# PROTOTYPICAL CALIBRATION FOR FEW-SHOT LEARNING OF LANGUAGE MODELS

**Zhixiong Han**,[*] **Yaru Hao, Li Dong, Yutao Sun**,[*] **Furu Wei**
Microsoft Research
{zhixhan8,sunyutao20001121}@gmail.com,
{yaruhao,lidong1,fuwei}@microsoft.com

## ABSTRACT

In-context learning of GPT-like models has been recognized as fragile across different hand-crafted templates, and demonstration permutations. In this work, we propose *prototypical calibration* to adaptively learn a more robust decision boundary for zero- and few-shot classification, instead of greedy decoding. Concretely, our method first adopts Gaussian mixture distribution to estimate the prototypical clusters for all categories. Then we assign each cluster to the corresponding label by solving a weighted bipartite matching problem. Given an example, its prediction is calibrated by the likelihood of prototypical clusters. Experimental results show that prototypical calibration yields a substantial improvement on a diverse set of tasks. Extensive analysis across different scales also indicates that our method calibrates the decision boundary as expected, greatly improving the robustness of GPT to templates, permutations, and class imbalance. The code will be released at https://github.com/zhixhan/ProCa.

## 1 INTRODUCTION

Large-scale language models (LMs) have shown strong generalization ability on a wide range of downstream tasks (Devlin et al., 2018; Radford et al., 2019; Yang et al., 2019; Lewis et al., 2019; Brown et al., 2020; Dong et al., 2019; Bao et al., 2020). Fine-tuning has been the common strategy to transfer the extensive knowledge to downstream tasks for a long time.

However, fine-tuning such large LMs suffers from the over-parameterization issue under few-shot settings. Brown et al. (2020) propose the concept of in-context learning with GPT, which enables LMs to quickly adapt to a new task by conditioning on hand-crafted prompts as shown in Figure 1. The prompts consist of task-specific templates and several input-label pairs (demonstrations). In-context learning is surprising as GPT can perform various tasks without any parameter updating.

It has been noticed that the predictions of GPT conditioned on prompts tend to bias toward some specific answers and can be highly volatile across different templates, demonstrations, and their permutations (Lu et al., 2021; Jiang et al., 2020). Zhao et al. (2021) propose to calibrate the model prediction by the content-free output to mitigate this problem. Rubin et al. (2021) and Lu et al. (2021) focus on the training examples retrieval and optimal ordering selection respectively to produce more performant prompts than random sampling. However, they did not explain why the in-context learning performance is fragile across different scenarios.

In this paper, we analyze the intrinsic reason for the instability of few-shot learning with GPT. We observe significant distinctions among the prediction distributions of GPT under different prompts. As shown in Figure 2, the conventional decision boundary of GPT (i.e., naively uses the output with the largest probability as the predicted label) often fails to discriminate the predictions. We argue that the predictions can be more discriminative when provided with a calibrated decision boundary.

Specifically, we term the model outputs of examples whose ground-truth are the same category as prototypical clusters and adopt Gaussian Mixture Model (GMM) to estimate the distributions of them for all categories. The decision boundaries of the prototypical clusters are adaptively learned,

---

[*] Contribution during internship at Microsoft Research.

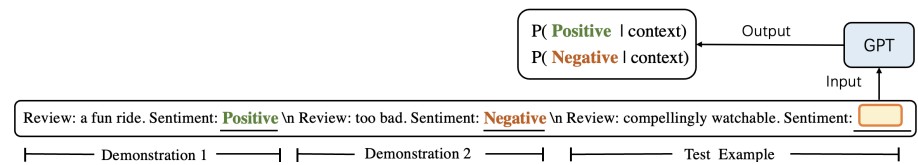

Figure 1: Example of few-shot learning with GPT.

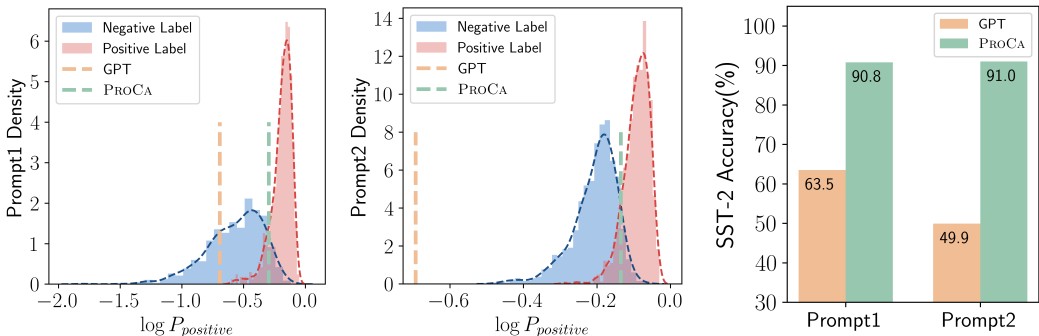

Figure 2: **Left** and **Middle**: Prediction distribution of GPT-2-XL under two different prompts for SST-2. Two distributions colored by blue and red represent model predictions for negative and positive ground-truth examples respectively. $P_{\text{positive}}$ denotes the prediction probability of positive label. The orange dashed line represents the decision boundary commonly used by GPT (i.e., $P_{\text{positive}} = 0.5$ for binary classification). The green dashed line represents the decision boundary of our prototypical calibration (PROCA). **Right**: Performance comparison of GPT-2-XL and PROCA under the two prompts, which indicates that PROCA is effective because the calibrated decision boundary is more discriminative for classification.

which is called prototypical calibration (PROCA). Then the prototypical clusters are assigned to the corresponding labels through weighted bipartite matching. We also propose to improve estimations according to cluster-label assignment. Finally, the predictions of test examples are more precise owing to the calibrated decision boundary (as shown in Figure 2).

Experimental results show that we achieve on average 13% absolute improvement for different sizes of GPT models across nine text classification datasets. We demonstrate that PROCA is effective across various templates and different demonstration permutations.

To summarize, our key contributions are as follows:

- We find that the decision boundary plays a critical role in few-shot evaluation. Moreover, performant decision boundaries are inconsistent across language models and prompts.

- We propose prototypical calibration to adaptively learn a better decision boundary for the few-shot classification of language models.

- Experiments show that PROCA achieves a 13% absolute improvement over the conventional approach on a wide range of text classification tasks.

## 2 DECISION BOUNDARY OF FEW-SHOT LEARNING WITH GPT

A decision boundary refers to an explicit prediction criterion in the output space for a given classification problem. As shown in Figure 2, two dashed lines represent two different decision boundaries, which classify examples into negative and positive categories. In this section, we explore the effect of the decision boundary on few-shot learning. We demonstrate that optimal decision boundaries are inconsistent under different LMs and prompts.

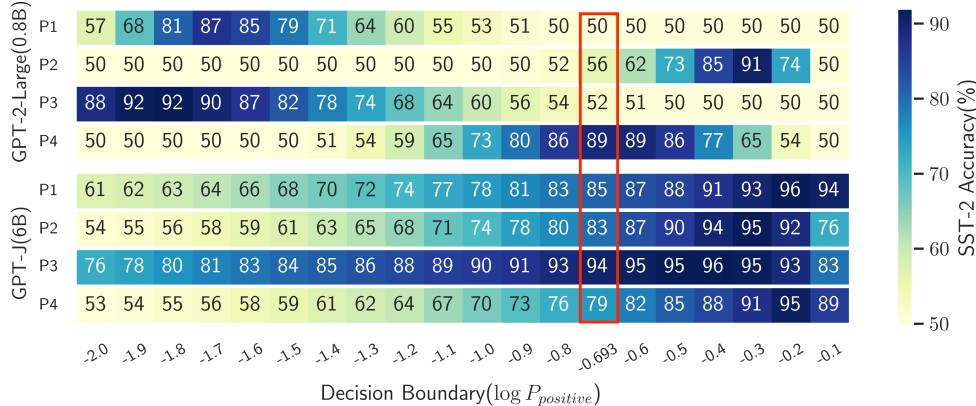

Figure 3: Few-shot performance of GPT-2-Large (0.8B) and GPT-J (6B) using different decision boundaries. P1, P2, P3, and P4 represent different prompts. The red rectangle indicates the performance under the conventional decision boundary ($P_{\text{positive}} = 0.5$ for the example task), i.e., naively using the outputs with larger probabilities as the predicted labels. It is observed that the decision boundary plays a critical role in few-shot evaluation.

**Decision boundary greatly influences the few-shot performance.** We evaluate the performance of different models and prompts using different decision boundaries. Results are shown in Figure 3. The red rectangle indicates the conventional decision boundary used by GPT, which naively decodes the label with a larger prediction probability. We observe that shifting the decision boundary can cause wild fluctuations of few-shot accuracy, from near state-of-the-art to random guessing. For each prompt, there is an exclusive region where the decision boundary is relatively robust. The model exhibits poor performance when the decision boundary is far from the robust region.

**Performant decision boundaries are not transferable across LMs or prompts.** Figure 3 demonstrates that all prompts exhibit strong performance if the decision boundary locates in the robust region. However, different prompts and models lead to different degrees of deviation between the optimal decision boundary and the conventional one. It suggests that performant decision boundaries are inconsistent across models or prompts. Based on the above analysis, we argue that all prompts can achieve better performance when the decision boundary is calibrated into the robust region.

## 3 PROTOTYPICAL CALIBRATION

We have illustrated that the conventional decision boundary generally deviates from the robust region, which renders in-context learning fragile. In this section, we present prototypical calibration (PROCA) to adaptively learn a better decision boundary.

### 3.1 PROTOTYPICAL CLUSTER ESTIMATION

Considering an $N$-way few-shot classification task, let $X$ denote the $N$-dimensional model outputs. For examples whose ground truth is the $n$-th category, the model outputs compose a prototypical cluster. For instance, the red and blue areas in Figure 2 refer to two prototypical clusters respectively.

We assume that each prototypical cluster follows a Gaussian distribution:

$$P_{\text{G}}(X|\boldsymbol{\mu_n}, \boldsymbol{\Sigma_n}) = \frac{1}{(2\pi)^{N/2}|\boldsymbol{\Sigma_n}|^{1/2}} \exp(-\frac{1}{2}(X - \boldsymbol{\mu_n})^T \boldsymbol{\Sigma_n}^{-1}(X - \boldsymbol{\mu_n})), \qquad (1)$$

where $\boldsymbol{\mu_n}$ and $\boldsymbol{\Sigma_n}$ are the mean vector, and covariance matrix of the distribution, respectively. Next, we estimate $N$ prototypical clusters for $N$ categories with Gaussian mixture model (GMM):

$$P_{\text{GMM}}(X) = \sum_{n=1}^{N} \alpha_n P_{\text{G}}(X|\boldsymbol{\mu_n}, \boldsymbol{\Sigma_n}), \qquad (2)$$

where $\alpha_n$ is the mixing coefficient for the $n$-th distribution. In our work, we formulate the model prediction $x = [x_1, x_2, ..., x_N]$ as follows:

$$x_n = \log \frac{\exp(o_n)}{\sum_{i=1}^{N} \exp(o_i)}, \tag{3}$$

where $o_n$ and $o_i$ are the logits predicted by GPT, corresponding to label token $n$ and label token $i$ respectively. Intuitively, $x_n$ represents the log probability of the $n$-th category.

After clarifying the GMM definition under few-shot learning, we utilize a small-scale unlabeled in-domain dataset, named as *estimate set* ($D_{\text{esti}}$), to estimate the parameters $\{\alpha_n, \boldsymbol{\mu_n}, \boldsymbol{\Sigma_n}\}_{n=1}^{N}$ by the Expectation-Maximization (EM) algorithm (Moon, 1996). Notice that the estimate set does not contain any human annotation. Specifically, EM is an iterative method to find the optimal estimation of GMM's parameters by maximizing the likelihood $\prod_{x \in D_{\text{esti}}} P_{\text{GMM}}(x)$.

## 3.2 CLUSTER-LABEL ASSIGNMENT

Then we assign the estimated prototypical clusters to the target labels. Concretely, for an estimation $e = \{(\alpha_n, \boldsymbol{\mu_n}, \boldsymbol{\Sigma_n})\}_{n=1}^{N}$, $\mu_{n,l}$ is the $l$-th element of $\mu_n$ and indicates how much the $n$-th cluster of $e$ belongs to label $l$. Therefore, we propose a cluster-label assignment score $\text{CLA}(\cdot)$, which represents the overall belongingness of a cluster-label assignment. Let the tuple $k = (k_1, k_2, \cdots, k_N)$ denote a cluster-label assignment, where $k$ is a permutation of $\{1, 2, ..., N\}$. It means that the $n$-th cluster is assigned to the label $k_n$. The assignment score $\text{CLA}(\cdot)$ is defined as:

$$\text{CLA}(e, k) = \sum_{n=1}^{N} \mu_{n,k_n}. \tag{4}$$

Then it is transformed into a weighted bipartite matching problem between $N$ clusters and $N$ labels. The optimal assignment is obtained by maximizing $\text{CLA}(e, k)$:

$$k^*(e) = \arg\max_{k \in \mathcal{K}} \text{CLA}(e, k), \tag{5}$$

where $\mathcal{K}$ indicates the set of all assignment permutations. In the worst case, this process requires $N!$ attempts to find the optimal assignment, which is time-consuming when $N$ is large, thus we adopt Kuhn-Munkres algorithm (Kuhn, 1955) to accelerate it.

## 3.3 ESTIMATION SELECTION BASED ON CLUSTER-LABEL ASSIGNMENT

The EM algorithm is empirically sensitive to the different initializations of GMM parameters. So we repeat the estimation multiple times with different random seeds. Then we define a metric to evaluate how good these estimations are and select the best estimation. As $\text{CLA}(e, k^*)$ reflects the overall label belongingness of the optimal assignment of an estimation $e$, thus it can be used to evaluate estimations. Formally, we select the estimation $e^*$ according to the assignment score of $k^*$ as follows:

$$e^* = \arg\max_{e \in \mathcal{E}} \text{CLA}(e, k^*(e)), \tag{6}$$

where $\mathcal{E}$ is the set of estimations obtained by different initializations of GMM parameters.

## 3.4 INFERENCE

After selecting the desired estimation $e^*$, we use GMM to make predictions instead of the conventional approach used in GPT (Brown et al., 2020). Due to the class-distribution discrepancy between the estimate set and the test set, we discard the mixing coefficient $\alpha_n$ of each sub-distribution during inference. For a test example, the LM prediction is $x$. It will be assigned to the most likely cluster:

$$\tilde{n} = \arg\max_{n=1, \cdots, N} P_{\text{G}}(x | \mu_n^*, \Sigma_n^*). \tag{7}$$

Finally, the predicted label is $k_{\tilde{n}}^*(e^*)$, where the cluster-label assignment $k^*(e^*)$ is obtained according to Equation (5).

# 4 EXPERIMENTS

## 4.1 EXPERIMENTAL SETUP

We evaluate five models from GPT-family including GPT-2-large (Radford et al., 2019) with 0.8B parameters, GPT-2-XL (Radford et al., 2019) with 1.5B parameters, GPT-neo (Black et al., 2021) with 2.7B parameters, GPT-J (Wang & Komatsuzaki, 2021) with 6B parameters, and Bloom (BigScience, 2022) with 176B parameters.

As for the estimate set, it can be constructed by generating from LMs (Lu et al., 2021; Wang et al., 2021; Meng et al., 2022; Ye et al., 2022) or sampling a light subset of training examples but without golden labels. For simplicity, we choose the latter way to construct the estimate set and we further compare their differences in Section 4.5. Moreover, the estimate set size is proportional to the number of classes of the task. For more details, please refer to Table 7 in Appendix.

We use the $k$-means algorithm to initialize GMM parameters to accelerate the convergence. The maximum iterations and the convergence threshold for each EM process are set to 100 and 1e-3 respectively. Moreover, we repeat the estimation multiple times with different random initializations to avoid getting stuck in local optima. It is worth noting that multiple repetitions bring little additional time consumption compared to the inference of GPT, we thus simply set it to 100 for all tasks.

## 4.2 EVALUATION PROTOCOL

We evaluate the proposed method on nine widely-used text-classification datasets including SST-2 (Socher et al., 2013), SST-5 (Socher et al., 2013), Subj (Pang & Lee, 2004), MR (Pang & Lee, 2005), AP (Zhang et al., 2015), DBPedia (Zhang et al., 2015), AGNews (Zhang et al., 2015), RTE (Dagan et al., 2005), and TREC (Voorhees & Tice, 2000). SST-2, SST-5, MR, and AP are sentiment classification tasks. RTE is a textual entailment recognition task and TREC is a text retrieval question classification task. Subj and AGNews are subjectivity and topic classification tasks respectively, and DBPeida is an ontology classification task. We use the full validation set for evaluation except for AGNews, DBPedia, and AP, for which we randomly sample 2000 test examples.

We compare PROCA with the conventional approach used by GPT (Brown et al., 2020) and contextual calibration (Zhao et al., 2021). Experiments are conducted under 0-shot, 1-shot, 4-shot, and 8-shot scenarios. We fix the template format for each dataset (details of templates are shown in Table 6) and use the randomly sampled training examples as demonstrations. We compute the average accuracy on the validation set over five random seeds for each setting except for Bloom using 2 seeds. We conduct the evaluation on 8 Tesla A100 GPUs for Bloom and Tesla V100 GPUs for other models.

## 4.3 MAIN RESULTS

We report the mean and standard deviation of accuracy across five different random seeds for GPT-2-XL, GPT-J, and Bloom in Table 1. The results of GPT-2-Large and GPT-neo are shown in Table 9 of Appendix. From Table 1 and Table 9, we observe that PROCA achieves, on average, a 13% absolute improvement compared to the conventional approach and a 6% absolute improvement over contextual calibration. In some cases, the absolute improvement can be up to 40% and 20% respectively, like GPT-J 0-shot on DBpedia and GPT-2-XL 8-shot on AGNews.

Results show that PROCA maintains high effectiveness across different model sizes and few-shot scenarios, indicating its strong generalization ability. Moreover, compared to the conventional approach, PROCA achieves considerable improvements with lower variance across different prompts in most cases, which suggests that PROCA can effectively calibrate the decision boundary for various prompts (as illustrated in Figure 2). It also reflects that our estimation strategy is reliable and insensitive to different estimate sets, because of the low variance of PROCA's zero-shot performance. We observe that the performance gain on Bloom is smaller than that on relatively small models. It suggests that huge LMs have less suffering on the decision boundary deviation problem. In addition, PROCA seems invalid for GPT-2-XL on RTE. We identify the reason is that the entailment recognition task is too challenging for relatively small models like GPT-2-XL and the output of LM on such

| Shot | Method | SST-2 | SST-5 | MR | Subj | AP | AGNews | DBpedia | RTE | TREC | Avg |
|---|---|---|---|---|---|---|---|---|---|---|---|
| | | | | | *GPT-2-XL 1.5B* | | | | | | |
| 0-shot | GPT | $58.7_{0.0}$ | $28.4_{0.0}$ | $58.9_{0.0}$ | $57.6_{0.0}$ | $\mathbf{51.8}_{0.0}$ | $41.6_{0.0}$ | $60.3_{0.0}$ | $50.0_{0.0}$ | $28.6_{0.0}$ | 48.4 |
| | ConCa | $69.3_{0.0}$ | $22.6_{0.0}$ | $66.9_{0.0}$ | $72.9_{0.0}$ | $49.8_{0.0}$ | $\mathbf{67.7}_{0.0}$ | $54.3_{0.0}$ | $\mathbf{50.4}_{0.0}$ | $\mathbf{42.8}_{0.0}$ | 55.2 |
| | PROCA | $\mathbf{84.8}_{0.2}$ | $\mathbf{45.0}_{1.3}$ | $\mathbf{82.0}_{0.2}$ | $\mathbf{73.3}_{0.1}$ | $49.8_{0.3}$ | $64.6_{1.4}$ | $\mathbf{73.6}_{3.0}$ | $49.2_{0.7}$ | $42.0_{2.7}$ | **62.7** |
| 1-shot | GPT | $59.8_{14.0}$ | $26.2_{8.5}$ | $51.3_{0.6}$ | $54.5_{8.6}$ | $51.0_{0.1}$ | $37.4_{6.7}$ | $51.3_{12.7}$ | $\mathbf{53.8}_{1.0}$ | $29.1_{6.5}$ | 46.0 |
| | ConCa | $76.4_{2.2}$ | $30.2_{5.7}$ | $69.4_{5.0}$ | $62.0_{7.0}$ | $60.3_{4.0}$ | $65.0_{3.8}$ | $70.9_{7.4}$ | $53.1_{0.9}$ | $40.5_{3.3}$ | 58.6 |
| | PROCA | $\mathbf{89.4}_{2.4}$ | $\mathbf{42.5}_{2.9}$ | $\mathbf{84.3}_{1.0}$ | $\mathbf{71.8}_{5.7}$ | $\mathbf{69.8}_{8.2}$ | $\mathbf{69.8}_{4.3}$ | $\mathbf{79.9}_{3.8}$ | $49.5_{1.9}$ | $\mathbf{43.6}_{5.0}$ | **66.7** |
| 4-shot | GPT | $66.3_{13.7}$ | $31.3_{7.4}$ | $56.5_{5.9}$ | $53.4_{4.9}$ | $50.9_{0.1}$ | $40.9_{13.0}$ | $61.3_{7.6}$ | $52.0_{3.5}$ | $23.8_{5.7}$ | 48.5 |
| | ConCa | $79.9_{10.2}$ | $33.5_{3.5}$ | $67.7_{8.9}$ | $68.0_{8.7}$ | $75.6_{5.9}$ | $59.9_{6.3}$ | $74.9_{5.0}$ | $\mathbf{52.9}_{0.7}$ | $41.1_{4.3}$ | 61.5 |
| | PROCA | $\mathbf{90.4}_{0.6}$ | $\mathbf{39.6}_{4.5}$ | $\mathbf{78.1}_{11.8}$ | $\mathbf{74.8}_{10.2}$ | $\mathbf{80.1}_{7.1}$ | $\mathbf{67.4}_{13.5}$ | $\mathbf{87.2}_{4.9}$ | $52.2_{1.5}$ | $\mathbf{46.0}_{2.5}$ | **68.4** |
| 8-shot | GPT | $57.0_{9.0}$ | $30.5_{7.9}$ | $65.2_{12.7}$ | $57.9_{11.2}$ | $50.9_{0.0}$ | $42.9_{4.2}$ | $67.9_{7.1}$ | $53.0_{2.1}$ | $37.2_{4.9}$ | 51.4 |
| | ConCa | $73.9_{11.6}$ | $28.7_{3.4}$ | $74.1_{8.4}$ | $68.3_{8.3}$ | $71.1_{7.4}$ | $55.9_{14.0}$ | $75.0_{4.2}$ | $\mathbf{53.1}_{0.2}$ | $45.8_{1.7}$ | 60.7 |
| | PROCA | $\mathbf{88.0}_{1.3}$ | $\mathbf{36.5}_{4.4}$ | $\mathbf{80.8}_{6.4}$ | $\mathbf{80.2}_{3.3}$ | $\mathbf{79.3}_{7.8}$ | $\mathbf{75.5}_{3.2}$ | $\mathbf{89.4}_{0.7}$ | $51.3_{2.0}$ | $\mathbf{46.0}_{2.5}$ | **69.7** |
| | | | | | *GPT-J 6B* | | | | | | |
| 0-shot | GPT | $66.6_{0.0}$ | $26.6_{0.0}$ | $65.9_{0.0}$ | $67.9_{0.0}$ | $54.2_{0.0}$ | $33.7_{0.0}$ | $21.8_{0.0}$ | $55.2_{0.0}$ | $23.4_{0.0}$ | 46.1 |
| | ConCa | $57.7_{0.0}$ | $35.4_{0.0}$ | $57.1_{0.0}$ | $59.9_{0.0}$ | $63.1_{0.0}$ | $\mathbf{60.1}_{0.0}$ | $49.9_{0.0}$ | $55.6_{0.0}$ | $42.2_{0.0}$ | 53.4 |
| | PROCA | $\mathbf{74.2}_{0.2}$ | $\mathbf{42.1}_{0.8}$ | $\mathbf{73.1}_{0.4}$ | $\mathbf{69.5}_{0.2}$ | $\mathbf{63.3}_{0.2}$ | $55.1_{0.4}$ | $\mathbf{66.1}_{1.5}$ | $\mathbf{57.0}_{1.0}$ | $\mathbf{53.4}_{6.1}$ | **61.5** |
| 1-shot | GPT | $67.7_{7.3}$ | $31.7_{4.9}$ | $68.1_{4.1}$ | $65.0_{10.9}$ | $92.9_{2.7}$ | $65.6_{14.6}$ | $65.6_{14.8}$ | $52.6_{4.6}$ | $41.8_{9.0}$ | 61.2 |
| | ConCa | $89.3_{2.2}$ | $46.5_{3.4}$ | $\mathbf{88.5}_{1.1}$ | $58.8_{3.0}$ | $93.5_{1.3}$ | $75.5_{5.7}$ | $79.9_{3.3}$ | $53.1_{0.8}$ | $\mathbf{64.7}_{5.3}$ | 72.2 |
| | PROCA | $\mathbf{90.8}_{1.7}$ | $\mathbf{47.6}_{2.5}$ | $87.9_{1.5}$ | $\mathbf{77.9}_{4.8}$ | $\mathbf{95.1}_{0.5}$ | $\mathbf{79.8}_{5.4}$ | $\mathbf{90.0}_{2.2}$ | $\mathbf{56.7}_{3.1}$ | $55.3_{6.4}$ | **75.7** |
| 4-shot | GPT | $88.6_{4.3}$ | $44.7_{3.3}$ | $84.4_{8.2}$ | $58.2_{6.3}$ | $89.4_{10.0}$ | $72.1_{6.5}$ | $80.5_{13.2}$ | $55.6_{6.7}$ | $38.1_{5.4}$ | 68.0 |
| | ConCa | $92.9_{3.7}$ | $\mathbf{47.7}_{4.4}$ | $87.8_{1.8}$ | $66.5_{11.7}$ | $93.4_{1.0}$ | $76.4_{4.0}$ | $88.6_{3.0}$ | $54.7_{1.5}$ | $48.5_{4.9}$ | 72.9 |
| | PROCA | $\mathbf{95.0}_{0.4}$ | $46.2_{4.6}$ | $\mathbf{89.4}_{1.9}$ | $\mathbf{79.4}_{5.8}$ | $\mathbf{95.8}_{0.8}$ | $\mathbf{79.9}_{6.6}$ | $\mathbf{91.9}_{2.6}$ | $\mathbf{61.2}_{2.7}$ | $\mathbf{57.1}_{5.3}$ | **77.3** |
| 8-shot | GPT | $91.1_{6.2}$ | $44.9_{2.9}$ | $89.5_{2.3}$ | $82.1_{3.9}$ | $95.2_{1.7}$ | $76.9_{9.7}$ | $87.7_{3.1}$ | $61.0_{3.9}$ | $44.5_{5.6}$ | 74.8 |
| | ConCa | $93.4_{1.8}$ | $46.6_{4.4}$ | $90.1_{0.5}$ | $80.5_{5.8}$ | $\mathbf{96.2}_{0.3}$ | $79.9_{6.4}$ | $90.8_{2.0}$ | $59.6_{4.8}$ | $53.5_{7.9}$ | 76.7 |
| | PROCA | $\mathbf{94.4}_{1.0}$ | $\mathbf{47.4}_{4.4}$ | $\mathbf{90.7}_{0.7}$ | $\mathbf{83.6}_{4.2}$ | $96.1_{0.5}$ | $\mathbf{84.2}_{1.8}$ | $\mathbf{95.1}_{0.5}$ | $\mathbf{61.7}_{7.2}$ | $\mathbf{61.0}_{7.6}$ | **79.4** |
| | | | | | *Bloom 176B* | | | | | | |
| 0-shot | Bloom | $73.4_{0.0}$ | $26.0_{0.0}$ | $71.0_{0.0}$ | $53.3_{0.0}$ | $60.1_{0.0}$ | $27.1_{0.0}$ | $48.5_{0.0}$ | $62.5_{0.0}$ | $\mathbf{59.0}_{0.0}$ | 53.4 |
| | ConCa | $73.9_{0.0}$ | $25.3_{0.0}$ | $71.8_{0.0}$ | $49.0_{0.0}$ | $51.1_{0.0}$ | $38.2_{0.0}$ | $61.0_{0.0}$ | $53.8_{0.0}$ | $41.0_{0.0}$ | 51.7 |
| | PROCA | $\mathbf{76.4}_{0.1}$ | $\mathbf{31.8}_{0.2}$ | $\mathbf{73.4}_{0.4}$ | $\mathbf{61.3}_{0.3}$ | $\mathbf{80.4}_{0.8}$ | $\mathbf{60.1}_{3.5}$ | $\mathbf{75.8}_{0.1}$ | $\mathbf{62.6}_{0.2}$ | $52.9_{0.5}$ | **63.9** |
| 1-shot | Bloom | $91.7_{2.6}$ | $31.1_{7.5}$ | $84.6_{2.3}$ | $60.4_{8.5}$ | $\mathbf{96.1}_{0.1}$ | $67.6_{0.9}$ | $81.8_{2.0}$ | $61.2_{3.4}$ | $55.1_{7.1}$ | 70.0 |
| | ConCa | $91.8_{1.6}$ | $38.9_{4.3}$ | $86.8_{1.6}$ | $51.2_{2.5}$ | $\mathbf{96.1}_{0.4}$ | $78.4_{0.5}$ | $80.4_{1.9}$ | $54.0_{5.6}$ | $\mathbf{69.3}_{1.3}$ | 71.9 |
| | PROCA | $\mathbf{93.6}_{0.6}$ | $\mathbf{47.5}_{2.8}$ | $\mathbf{88.0}_{0.8}$ | $\mathbf{72.0}_{1.8}$ | $95.7_{0.4}$ | $\mathbf{81.6}_{0.7}$ | $\mathbf{83.7}_{1.8}$ | $\mathbf{65.7}_{0.4}$ | $67.5_{2.5}$ | **77.3** |
| 4-shot | Bloom | $\mathbf{96.3}_{0.1}$ | $46.7_{0.8}$ | $87.3_{5.3}$ | $72.2_{6.4}$ | $94.2_{2.5}$ | $68.8_{3.2}$ | $86.2_{1.4}$ | $64.1_{2.4}$ | $29.1_{0.9}$ | 71.7 |
| | ConCa | $96.0_{0.1}$ | $46.9_{2.9}$ | $89.7_{1.1}$ | $70.4_{7.7}$ | $94.2_{1.9}$ | $78.0_{0.1}$ | $86.6_{2.4}$ | $56.3_{0.7}$ | $\mathbf{64.8}_{7.6}$ | 75.9 |
| | PROCA | $95.7_{0.2}$ | $\mathbf{50.2}_{2.6}$ | $\mathbf{91.2}_{0.1}$ | $\mathbf{78.5}_{0.5}$ | $\mathbf{95.8}_{0.5}$ | $\mathbf{82.7}_{1.2}$ | $\mathbf{87.0}_{1.3}$ | $\mathbf{68.6}_{0.4}$ | $56.8_{4.8}$ | **78.5** |
| 8-shot | Bloom | $94.6_{2.0}$ | $43.2_{3.5}$ | $90.9_{0.8}$ | $78.6_{2.2}$ | $\mathbf{96.0}_{0.9}$ | $75.4_{1.9}$ | $88.4_{2.1}$ | $65.9_{2.4}$ | $48.9_{6.7}$ | 75.8 |
| | ConCa | $\mathbf{96.1}_{0.2}$ | $42.2_{5.5}$ | $91.0_{0.9}$ | $75.8_{1.7}$ | $95.9_{0.4}$ | $81.9_{2.0}$ | $\mathbf{89.5}_{2.6}$ | $59.0_{0.5}$ | $\mathbf{73.9}_{1.1}$ | 78.4 |
| | PROCA | $95.3_{1.3}$ | $\mathbf{53.1}_{1.6}$ | $\mathbf{92.0}_{0.6}$ | $\mathbf{80.6}_{1.9}$ | $95.6_{0.8}$ | $\mathbf{82.1}_{2.0}$ | $85.1_{3.7}$ | $\mathbf{69.5}_{2.7}$ | $68.6_{7.8}$ | **80.2** |

Table 1: Performance comparisons among the conventional approach (GPT; Brown et al. 2020), contextual calibration (ConCa; Zhao et al. 2021) and prototypical calibration (PROCA; *Ours*). We report the mean and the standard deviation of accuracy across 5 different prompts on the validation set except for Bloom, for which we only use 2 random seeds to reduce the computational cost. We also show the average performance across nine datasets. The results of ConCa are replicated based on the released code[1]. The standard deviation of 0-shot accuracy for PROCA is caused by the difference of estimate sets over 5 random seeds. It shows that PROCA generally outperforms GPT and ConCa.

challenging tasks is no more discriminative (same for GPT-2-Large, as shown in Table 9 in Appendix).

## 4.4 EFFECTIVENESS ANALYSIS

We conduct more experiments to verify the effectiveness of PROCA. The experimental results are the average accuracy of GPT-2-XL conditioned on 5 different 4-shot prompts unless otherwise specified.

**PROCA is consistently effective across different templates.** We conduct the experiments across nine different prompts templates and label spaces (details of templates are shown in Table 8 of

---

[1] https://www.github.com/tonyzhaozh/few-shot-learning

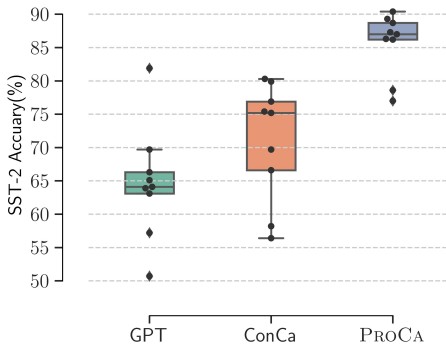

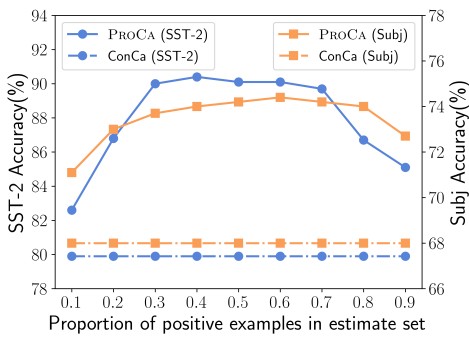

Figure 4: Performance comparison across nine different templates.

Figure 5: The impact of class-imbalanced estimate set on PROCA's performance.

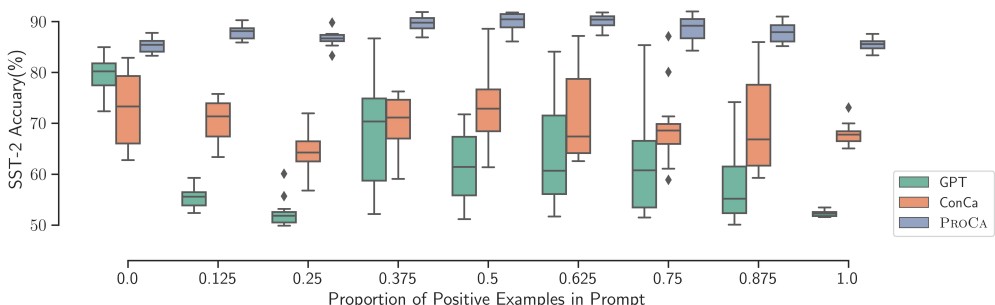

Figure 6: Performance comparison under different label proportions and permutations of demonstrations. Each box indicates the accuracy of twelve randomly sampled permutations.

Appendix). The performance comparison among three approaches on SST-2 is shown in Figure 4. We observe that contextual calibration remains high variance although it improves the average accuracy. However, our proposed prototypical calibration can bring a large improvement with low variance, which indicates that PROCA is effective on various prompt templates.

**PROCA is robust under demonstration perturbations.** Previous works (Zhao et al., 2021; Lu et al., 2021) have noticed that the order of training examples has significant effects on the performance of few-shot demonstrations. In this part, we evaluate our prototypical calibration conditioned on nine 8-shot prompts with different class proportions for SST-2, and show the accuracy of twelve randomly sampled orderings for each proportion in Figure 6. We find that contextual calibration can improve the performance in most cases but is still sensitive to the orderings. However, PROCA is significantly superior to the others and keeps an extremely low variance across different permutations, indicating the non-sensitiveness to the class proportion and permutation. It is also shown that although the class-balanced prompts tend to have higher performance, there are some exceptions(e.g., the prompt with all negative samples is the most performant one for both the conventional approach and contextual calibration). We think that it is GPT-2-XL's intrinsic bias to the positive class that leads toward counter-intuitive results.

**PROCA is robust to class imbalance.** Due to the unavailability of the labels of the estimate examples, PROCA may suffer the problem of class imbalance. We construct nine estimate sets with different imbalance levels for SST-2 and Subj by controlling the proportion of positive examples in the sampled set. Then we evaluate PROCA and contextual calibration on them. The experimental results in Figure 5 show that the estimate set's class imbalance level affects the performance of PROCA to some extent and the class-balanced estimate set can lead to higher accuracy. As described in Section 3.1, standard GMM estimates the weight of each cluster, which reflects the proportion of different classes in the estimate set. Owing to our "weights-cutting" operation, the problem

of class imbalance has a much less negative impact on PROCA, which even with an extremely class-imbalanced estimate set surpasses contextual calibration on both SST-2 and Subj. Besides, the absolute advancement for the class-balanced estimate set can reach 20% and 15% respectively.

| Shot | Method | SST-2 | SST-5 | MR | Subj | AP | AGNews | DBpedia | RTE | TREC |
|------|--------|-------|-------|-----|------|-----|--------|---------|-----|------|
| 4-shot | GPT-neo | $84.5_{8.7}$ | $33.2_{8.0}$ | $68.7_{13.4}$ | $61.9_{14.8}$ | $85.8_{10.3}$ | $68.2_{5.7}$ | $71.8_{10.9}$ | $47.9_{0.8}$ | $37.2_{6.4}$ |
|  | ConCa | $91.7_{1.1}$ | $41.3_{5.0}$ | $81.0_{7.2}$ | $62.6_{11.5}$ | $\mathbf{93.4}_{0.7}$ | $60.4_{8.9}$ | $86.7_{3.5}$ | $52.9_{3.6}$ | $57.3_{8.5}$ |
|  | PROCA-g | $\mathbf{91.9}_{1.4}$ | $\mathbf{43.5}_{3.4}$ | $83.9_{4.3}$ | $73.4_{6.9}$ | $91.6_{0.8}$ | $70.7_{5.4}$ | $80.1_{2.9}$ | $50.4_{1.6}$ | $57.3_{2.5}$ |
|  | PROCA-t | $91.6_{1.8}$ | $38.7_{5.6}$ | $\mathbf{85.6}_{1.2}$ | $\mathbf{79.7}_{2.9}$ | $93.3_{0.5}$ | $\mathbf{75.6}_{5.6}$ | $\mathbf{90.4}_{3.7}$ | $\mathbf{55.0}_{1.0}$ | $\mathbf{59.2}_{2.9}$ |
| 8-shot | GPT-neo | $68.0_{19.2}$ | $31.3_{6.9}$ | $70.2_{14.4}$ | $57.5_{8.1}$ | $90.4_{2.7}$ | $66.5_{8.0}$ | $78.8_{6.3}$ | $49.2_{2.6}$ | $50.8_{5.6}$ |
|  | ConCa | $81.2_{9.1}$ | $33.9_{4.7}$ | $77.8_{9.6}$ | $71.0_{5.7}$ | $93.6_{0.9}$ | $73.4_{3.5}$ | $90.3_{1.0}$ | $51.3_{6.6}$ | $56.0_{8.0}$ |
|  | PROCA-g | $90.1_{2.4}$ | $\mathbf{39.6}_{4.1}$ | $\mathbf{81.1}_{4.5}$ | $75.5_{3.8}$ | $92.0_{1.4}$ | $77.3_{4.5}$ | $81.8_{2.4}$ | $52.4_{2.7}$ | $\mathbf{63.4}_{5.2}$ |
|  | PROCA-t | $\mathbf{91.9}_{1.2}$ | $39.4_{4.0}$ | $77.8_{13.9}$ | $\mathbf{81.3}_{3.8}$ | $\mathbf{93.9}_{0.7}$ | $\mathbf{78.9}_{2.5}$ | $\mathbf{92.0}_{1.5}$ | $\mathbf{56.8}_{1.8}$ | $56.0_{3.6}$ |

Table 2: 4- and 8-shot performance comparison of different estimate set construction methods for GPT-neo across nine text classification tasks. PROCA-g and PROCA-t represent PROCA based on the unlabeled estimate set generated by LM and randomly sampled from the training set, respectively.

## 4.5 ABLATION STUDIES

**Comparison between different estimate set construction methods.** There are two ways to construct the estimate set. One is using light unlabeled examples from the training set, which is simple and convenient. The other is utilizing the generation ability of LMs to construct unlabeled dataset (Lu et al., 2021; Wang et al., 2021; Meng et al., 2022; Ye et al., 2022). For the generation method, we follow Lu et al. (2021) to generate diverse estimate examples based on the various permutations of demonstrations. Specifically, we only use two labeled examples per category as demonstrations for generation except for DBPedia(one labeled example per category) and all these labeled examples are not involved in the evaluation. The 4-shot and 8-shot experimental results are shown in Table 2 and the 0-shot and 1-shot results are shown in Table 5 of Appendix. We observe that PROCA greatly outperforms the original LM whether using unlabeled data generated by LM or randomly sampled from the training set. It also shows that PROCA-t performs slightly better than PROCA-g and we speculate that it is due to the lower quality of the unlabeled data generated by LM.

**A relatively small-scale estimate set is sufficient for PROCA.** In Figure 7, we evaluate PROCA with ten different estimate set sizes across five datasets. We report the average accuracy over five randomly sampled estimate sets, conditioning on the same 4-shot prompt in each setting. We observe that increasing the scale of the estimate set within a certain small range can greatly improve the classification accuracy and reduce the variance. However, a larger estimate set can hardly bring further improvement, which indicates that a small estimate set can support PROCA to be optimal. It also shows that PROCA has acceptable performances with just several estimate examples on SST-2, MR, and Subj, which even surpasses both the conventional approach and contextual calibration.

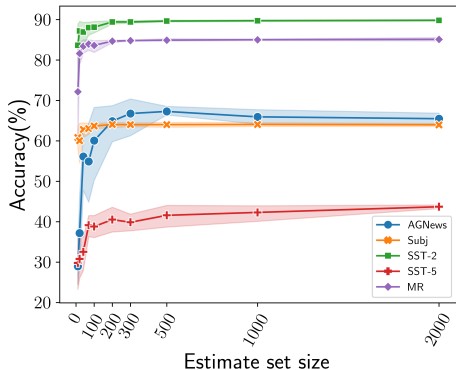

Figure 7: Performance of PROCA across different estimate set sizes.

**Estimation selection according to assignment score is useful to PROCA.** The standard estimation of GMM aims to maximize the likelihood of all observations and select the optimal estimated parameters among multiple repetitions. We argue that the estimation with maximum likelihood is not consistently beneficial to PROCA especially in multi-classes tasks, because there is no supervision to force the predictions to be assigned to their inclined classes during the estimation procedure. We propose to determine estimation according to the assignment score (as described in Section 3.3). We compare PROCA with the two strategies for GPT-2-XL and GPT-J respectively in Table 3. It indicates that our determination strategy can achieve more stable improvements on AGNnews and DBPedia regardless of model size.

| Dataset | Strategy | 0-shot | | 1-shot | | 4-shot | | 8-shot | |
|---------|----------|--------|--------|--------|--------|--------|--------|--------|--------|
| | | GPT-2 | GPT-J | GPT-2 | GPT-J | GPT-2 | GPT-J | GPT-2 | GPT-J |
| AGNews | max-likelihood | $64.0_{1.1}$ | $\mathbf{55.2}_{0.3}$ | $65.8_{5.2}$ | $79.2_{5.8}$ | $60.1_{11.1}$ | $79.7_{6.7}$ | $71.7_{8.7}$ | $78.1_{11.7}$ |
| | assignment score | $\mathbf{64.6}_{1.4}$ | $55.1_{0.4}$ | $\mathbf{69.8}_{4.3}$ | $\mathbf{79.8}_{5.4}$ | $\mathbf{67.4}_{13.5}$ | $\mathbf{79.9}_{6.6}$ | $\mathbf{75.5}_{3.2}$ | $\mathbf{84.2}_{1.8}$ |
| DBPeida | max-likelihood | $63.8_{7.5}$ | $59.2_{4.6}$ | $71.8_{7.3}$ | $77.7_{3.8}$ | $76.1_{5.5}$ | $82.2_{3.9}$ | $79.3_{5.0}$ | $83.8_{2.4}$ |
| | assignment score | $\mathbf{73.6}_{3.0}$ | $\mathbf{66.1}_{1.5}$ | $\mathbf{79.9}_{3.8}$ | $\mathbf{90.0}_{2.2}$ | $\mathbf{87.2}_{4.9}$ | $\mathbf{91.9}_{2.6}$ | $\mathbf{89.4}_{0.7}$ | $\mathbf{95.1}_{0.5}$ |

Table 3: Performance of PROCA with different strategies of estimation selection (maximum likelihood, and assignment score as in Equation (4)) for GPT-2-XL and GPT-J on AGNews and DBPedia.

## 5 RELATED WORK

**Instability of Few-shot Learning with Language Models.** It has been recognized that the few-shot performance of language models is unstable under different in-context scenarios. Language models are prone to predict some specific labels due to the intrinsic bias or demonstration permutations (Zhao et al., 2021; Lu et al., 2021). Lu et al. (2021) demonstrate LM's sensitivity to the order of few-shot demonstrations, and introduced an Entropy-based metric to select the most performant prompts. Zhao et al. (2021) attribute the instability to three biases of prompts, including majority bias, recency bias, and common token bias, and proposed a contextual calibration approach. However, the selected content-free test inputs can not precisely reflect the bias of models and lead to the problem of over-correction or under-correction. On the contrary, we adaptively provide the classification criterion according to the text inputs' overall prediction distribution, and completely calibrate the bias introduced by models and prompts.

**Improving and Understanding In-context Learning with Language Models.** Due to the instability, prior efforts propose various methods to improve the in-context learning performance. Holtzman et al. (2021) explores the surface form competition problem in zero-shot models and proposes Domain Conditional Pointwise Mutual Information to reweigh the answer scores. Min et al. (2021) introduces a noisy channel approach that computes the conditional probability of the input given the label, which boosts improvements with lower variance and higher worst-case accuracy. Moreover, Liu et al. (2021) focus on prompt engineering to construct more semantically-similar demonstrations. To the best of our knowledge, we are the first to study the intrinsic reason for the instability of in-context learning from the perspective of the decision boundary and propose prototypical calibration to improve it. Another line of work aims to understand how in-context learning works through casting it as implicit Bayesian inference (Xie et al., 2022), analyzing corpora sources and statistics (Shin et al., 2022; Razeghi et al., 2022) and learning which aspects of demonstrations contribute most to the downstream performance (Min et al., 2022).

## 6 CONCLUSION AND LIMITATION

To our analysis, the decision boundary is of critical importance to the performance of few-shot demonstrations and the traditional decision boundary leads to the fragility of prompting LMs. We propose prototypical calibration to adaptively learn a more robust decision boundary. Experiments show that the calibrated decision boundary is effective across various prompt templates, class proportions, and permutations. We achieve on average a 13% absolute improvement across different sizes of pretrained language models on nine popular text classification tasks.

A limitation of our method is that it is not applicable for tasks whose label space is open-ended since a fixed label space is necessary for estimating prototypical clusters. Furthermore, our method is designed for in-context learning on individual downstream tasks, it fails to calibrate the inherent bias of language models like gender and occupation bias. For future work, we would like to extend our method to the tasks with open-ended answer space, such as generative question-answering and text summarization tasks.

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

## A EXPERIMENTAL DETAILS AND ADDITIONAL RESULTS

Table 4: Average performance of the conventional approach, contextual calibration (ConCa) and PROCA for GPT-2-Large and GPT-neo across nine text classification tasks.

| Shot | Method | SST-2 | SST-5 | MR | Subj | AP | AGNews | DBpedia | RTE | TREC | Avg |
|---|---|---|---|---|---|---|---|---|---|---|---|
| | | | | | *GPT-2-Large 0.8B* | | | | | | |
| 0-shot | GPT | $72.1_{0.0}$ | $26.2_{0.0}$ | $70.2_{0.0}$ | $51.7_{0.0}$ | $53.2_{0.0}$ | $33.5_{0.0}$ | $56.7_{0.0}$ | $52.7_{0.0}$ | $34.6_{0.0}$ | 50.1 |
| | ConCa | $80.5_{0.0}$ | $41.7_{0.0}$ | $66.9_{0.0}$ | $54.5_{0.0}$ | $60.6_{0.0}$ | $59.3_{0.0}$ | $57.4_{0.0}$ | $53.4_{0.0}$ | $32.0_{0.0}$ | 56.3 |
| | PROCA | $\mathbf{85.6}_{0.3}$ | $\mathbf{43.5}_{0.8}$ | $\mathbf{82.4}_{0.3}$ | $\mathbf{69.5}_{0.5}$ | $\mathbf{61.2}_{0.6}$ | $\mathbf{59.6}_{2.2}$ | $\mathbf{79.8}_{1.5}$ | $\mathbf{53.6}_{0.4}$ | $\mathbf{48.3}_{1.4}$ | **64.8** |
| 1-shot | GPT | $56.1_{10.8}$ | $28.5_{10.4}$ | $53.3_{4.4}$ | $50.5_{1.5}$ | $50.7_{0.8}$ | $28.8_{5.3}$ | $38.3_{14.3}$ | $\mathbf{52.8}_{0.4}$ | $32.7_{3.4}$ | 43.5 |
| | ConCa | $75.3_{12.7}$ | $\mathbf{40.1}_{3.6}$ | $68.7_{12.0}$ | $61.4_{2.2}$ | $56.1_{6.4}$ | $\mathbf{65.0}_{5.8}$ | $69.8_{9.2}$ | $51.3_{3.4}$ | $39.6_{8.1}$ | 58.6 |
| | PROCA | $\mathbf{83.1}_{2.1}$ | $37.9_{2.5}$ | $\mathbf{79.5}_{1.0}$ | $\mathbf{65.2}_{3.5}$ | $\mathbf{84.1}_{2.0}$ | $53.7_{9.8}$ | $\mathbf{78.8}_{12.4}$ | $51.3_{2.5}$ | $\mathbf{47.2}_{7.5}$ | **64.5** |
| 4-shot | GPT | $52.7_{2.0}$ | $31.5_{8.5}$ | $59.4_{8.5}$ | $60.3_{10.7}$ | $52.8_{2.7}$ | $33.7_{5.3}$ | $38.6_{16.8}$ | $51.0_{2.7}$ | $32.4_{6.9}$ | 45.8 |
| | ConCa | $71.8_{11.9}$ | $\mathbf{42.3}_{2.3}$ | $70.1_{15.3}$ | $58.5_{10.3}$ | $72.0_{7.2}$ | $56.6_{5.3}$ | $77.6_{4.1}$ | $\mathbf{52.9}_{2.1}$ | $42.2_{7.1}$ | 60.4 |
| | PROCA | $\mathbf{86.9}_{3.0}$ | $35.5_{4.6}$ | $\mathbf{81.5}_{4.2}$ | $\mathbf{75.1}_{3.2}$ | $\mathbf{87.6}_{3.9}$ | $\mathbf{60.3}_{11.0}$ | $\mathbf{82.7}_{6.1}$ | $52.2_{2.5}$ | $\mathbf{49.4}_{6.2}$ | **67.9** |
| 8-shot | GPT | $72.0_{17.4}$ | $31.2_{7.0}$ | $61.8_{9.9}$ | $57.9_{8.7}$ | $51.8_{1.9}$ | $40.9_{6.7}$ | $60.4_{12.7}$ | $\mathbf{53.9}_{1.4}$ | $38.0_{6.0}$ | 52.0 |
| | ConCa | $83.8_{12.0}$ | $\mathbf{40.0}_{5.5}$ | $72.7_{7.6}$ | $63.4_{11.0}$ | $62.6_{6.9}$ | $52.6_{10.8}$ | $77.4_{3.6}$ | $49.6_{2.3}$ | $47.0_{7.7}$ | 61.0 |
| | PROCA | $\mathbf{88.0}_{3.3}$ | $37.6_{4.3}$ | $\mathbf{84.5}_{0.6}$ | $\mathbf{79.7}_{5.7}$ | $\mathbf{90.5}_{2.4}$ | $\mathbf{72.5}_{2.6}$ | $\mathbf{87.9}_{0.7}$ | $52.3_{3.2}$ | $\mathbf{47.5}_{5.0}$ | **71.2** |
| | | | | | *GPT-neo 2.7B* | | | | | | |
| 0-shot | GPT | $59.6_{0.0}$ | $27.6_{0.0}$ | $58.0_{0.0}$ | $\mathbf{77.4}_{0.0}$ | $\mathbf{48.6}_{0.0}$ | $45.0_{0.0}$ | $35.1_{0.0}$ | $\mathbf{54.5}_{0.0}$ | $22.6_{0.0}$ | 47.6 |
| | ConCa | $77.0_{0.0}$ | $29.0_{0.0}$ | $72.5_{0.0}$ | $52.4_{0.0}$ | $48.4_{0.0}$ | $61.3_{0.0}$ | $56.6_{0.0}$ | $52.0_{0.0}$ | $29.6_{0.0}$ | 53.2 |
| | PROCA | $\mathbf{86.1}_{0.7}$ | $\mathbf{45.5}_{1.4}$ | $\mathbf{80.8}_{0.1}$ | $77.2_{0.4}$ | $46.1_{0.1}$ | $\mathbf{61.5}_{1.0}$ | $\mathbf{71.3}_{1.9}$ | $54.3_{0.9}$ | $\mathbf{41.8}_{4.3}$ | **62.7** |
| 1-shot | GPT | $73.1_{6.3}$ | $33.4_{9.3}$ | $72.3_{6.2}$ | $53.2_{6.6}$ | $80.3_{4.5}$ | $45.2_{10.2}$ | $54.3_{12.3}$ | $47.0_{0.5}$ | $29.6_{13.1}$ | 54.3 |
| | ConCa | $84.0_{11.5}$ | $40.1_{2.7}$ | $83.7_{4.4}$ | $56.5_{6.4}$ | $82.0_{3.1}$ | $58.7_{3.7}$ | $73.6_{4.0}$ | $48.9_{2.4}$ | $55.2_{7.7}$ | 64.7 |
| | PROCA | $\mathbf{90.0}_{2.0}$ | $\mathbf{41.2}_{4.4}$ | $\mathbf{86.2}_{0.7}$ | $\mathbf{68.2}_{3.3}$ | $\mathbf{86.5}_{2.1}$ | $\mathbf{69.8}_{3.4}$ | $\mathbf{85.5}_{3.6}$ | $\mathbf{52.9}_{2.0}$ | $\mathbf{57.6}_{4.5}$ | **70.9** |
| 4-shot | GPT | $84.5_{8.7}$ | $33.2_{8.0}$ | $68.7_{13.4}$ | $61.9_{14.8}$ | $85.8_{10.3}$ | $68.2_{5.7}$ | $71.8_{10.9}$ | $47.9_{0.8}$ | $37.2_{6.4}$ | 62.1 |
| | ConCa | $\mathbf{91.7}_{1.1}$ | $\mathbf{41.3}_{5.0}$ | $81.0_{7.2}$ | $62.6_{11.5}$ | $\mathbf{93.4}_{0.7}$ | $60.4_{8.9}$ | $86.7_{3.5}$ | $52.9_{3.6}$ | $57.3_{8.5}$ | 69.7 |
| | PROCA | $91.6_{1.8}$ | $38.7_{5.6}$ | $\mathbf{85.6}_{1.2}$ | $\mathbf{79.7}_{2.9}$ | $93.3_{0.5}$ | $\mathbf{75.6}_{5.6}$ | $\mathbf{90.4}_{3.7}$ | $\mathbf{55.0}_{1.0}$ | $\mathbf{59.2}_{2.9}$ | **74.3** |
| 8-shot | GPT | $68.0_{19.2}$ | $31.3_{6.9}$ | $70.2_{14.4}$ | $57.5_{8.1}$ | $90.4_{2.7}$ | $66.5_{8.0}$ | $78.8_{6.3}$ | $49.2_{2.6}$ | $50.8_{5.6}$ | 62.5 |
| | ConCa | $81.2_{9.1}$ | $33.9_{4.7}$ | $\mathbf{77.8}_{9.6}$ | $71.0_{5.7}$ | $93.6_{0.9}$ | $73.4_{3.5}$ | $90.3_{1.0}$ | $51.3_{6.6}$ | $\mathbf{56.0}_{8.0}$ | 69.8 |
| | PROCA | $\mathbf{91.9}_{1.2}$ | $\mathbf{39.4}_{4.0}$ | $77.8_{13.9}$ | $\mathbf{81.3}_{3.8}$ | $\mathbf{93.9}_{0.7}$ | $\mathbf{78.9}_{2.5}$ | $\mathbf{92.0}_{1.5}$ | $\mathbf{56.8}_{1.8}$ | $56.0_{3.6}$ | **74.2** |

Table 5: 0-shot and 1-shot performance comparison of different estimate set construction methods for GPT-neo across nine text classification tasks. PROCA-g and PROCA-t represent PROCA based on the unlabeled estimate set generated by LM and randomly sampled from the training set, respectively.

| Shot | Method | SST-2 | SST-5 | MR | Subj | AP | AGNews | DBpedia | RTE | TREC |
|---|---|---|---|---|---|---|---|---|---|---|
| 0-shot | GPT-neo | $59.6_{0.0}$ | $27.6_{0.0}$ | $58.0_{0.0}$ | $\mathbf{77.4}_{0.0}$ | $\mathbf{48.6}_{0.0}$ | $45.0_{0.0}$ | $35.1_{0.0}$ | $\mathbf{54.5}_{0.0}$ | $22.6_{0.0}$ |
| | ConCa | $77.0_{0.0}$ | $29.0_{0.0}$ | $72.5_{0.0}$ | $52.4_{0.0}$ | $48.4_{0.0}$ | $61.3_{0.0}$ | $56.6_{0.0}$ | $52.0_{0.0}$ | $29.6_{0.0}$ |
| | PROCA-g | $85.9_{0.0}$ | $43.9_{0.0}$ | $80.7_{0.0}$ | $71.9_{0.0}$ | $46.6_{0.0}$ | $\mathbf{62.7}_{0.0}$ | $42.9_{0.0}$ | $53.4_{0.0}$ | $27.4_{0.0}$ |
| | PROCA-t | $\mathbf{86.1}_{0.7}$ | $\mathbf{45.5}_{1.4}$ | $\mathbf{80.8}_{0.1}$ | $77.2_{0.4}$ | $46.1_{0.1}$ | $61.5_{1.0}$ | $\mathbf{71.3}_{1.9}$ | $54.3_{0.9}$ | $\mathbf{41.8}_{4.3}$ |
| 1-shot | GPT-neo | $73.1_{6.3}$ | $33.4_{9.3}$ | $72.3_{6.2}$ | $53.2_{6.6}$ | $80.3_{4.5}$ | $45.2_{10.2}$ | $54.3_{12.3}$ | $47.0_{0.5}$ | $29.6_{13.1}$ |
| | ConCa | $84.0_{11.5}$ | $40.1_{2.7}$ | $83.7_{4.4}$ | $56.5_{6.4}$ | $82.0_{3.1}$ | $58.7_{3.7}$ | $73.6_{4.0}$ | $48.9_{2.4}$ | $55.2_{7.7}$ |
| | PROCA-g | $89.0_{2.1}$ | $37.3_{5.2}$ | $82.0_{2.8}$ | $66.9_{2.9}$ | $85.6_{1.9}$ | $67.3_{3.0}$ | $74.1_{5.2}$ | $49.8_{2.3}$ | $48.6_{5.3}$ |
| | PROCA-t | $\mathbf{90.0}_{2.0}$ | $\mathbf{41.2}_{4.4}$ | $\mathbf{86.2}_{0.7}$ | $\mathbf{68.2}_{3.3}$ | $\mathbf{86.5}_{2.1}$ | $\mathbf{69.8}_{3.4}$ | $\mathbf{85.5}_{3.6}$ | $\mathbf{52.9}_{2.0}$ | $\mathbf{57.6}_{4.5}$ |

Table 6: Prompt templates for nine datasets in our experiments.

| Dataset | Template | Label Space |
|---|---|---|
| SST-2 | Review: {Sentence}
Sentiment: {Label} | Positive / Negative |
| SST-5 | Review: {Sentence}
Sentiment: {Label} | terrible / bad / okay / good / great |
| MR | Review: {Sentence}
Sentiment: {Label} | Positive / Negative |
| Subj | Input: {Sentence}
Type: {Label} | objective / subjective |
| AP | Title: {Title}
Review: {Content}
Is the review positive or negative? {Label} | Positive / Negative |
| AGNews | Classify the news articles into the categories of World, Sports, Business, and Technology.

Article: {Sentence}
Answer: {Label} | World / Sports / Business / Technology |
| DBPedia | Classify the documents based on whether they are about a Company, School, Artist, Athlete, Politician, Transportation, Building, Nature, Village, Animal, Plant, Album, Film, or Book.

Article: {Sentence}
Answer: {Label} | Company / School / Artist / Athlete / Politician / Transportation / Building / Nature / Village / Animal / Plant / Album / Film / Book |
| RTE | {Premise}
question: {Hypothesis} True or False?
answer: {Label} | False / True |
| TREC | Classify the questions based on whether their answer type is a Number, Location, Person, Description, Entity, or Abbreviation.

Question: {Sentence}
Answer Type: {Label} | Number / Location / Person / Description / Entity / Abbreviation |

Table 7: Estimate set size for nine datasets in our experiments. GPTs include GPT-2-Large, GPT-2-XL, GPT-neo, and GPT-J.

| LM | SST-2 | SST-5 | MR | Subj | AP | AGNews | DBpedia | RTE | TREC |
|---|---|---|---|---|---|---|---|---|---|
| GPT | 500 | 2000 | 1000 | 1000 | 1000 | 2000 | 3000 | 1000 | 2000 |
| Bloom | 300 | 1000 | 500 | 500 | 500 | 1000 | 2000 | 500 | 1000 |

## B  COMPARISONS AMONG USING LOGITS, PROBABILITY, AND LOG-PROBABILITY FOR PROCA.

We evaluate the performance of PROCA based on three commonly used model output formalizations including logits, probability, and log-probability on SST-2, AGNews, and AP for GPT-2-XL. The experimental results are plotted in Figure 8. We find that PROCA with log-probability consistently outperforms that with probability, especially for AGNews. We identify that the log operation makes the predictions more fitted to Gaussian distribution and more discrepant. Although PROCA with logits seems to have superior performance when few-shot demonstrations are provided, it degrades severely at zero-shot setting. It suggests that output embedding normalization over label space is necessary to PROCA, because the output is more likely biased to the tokens outside the label space when no prompts are accessed, which is consistent with the conclusion of Min et al. (2022).

Table 8: Different templates used in Section 4.4 for SST-2.

| ID | Template | Label Space |
|---|---|---|
| 1 | Review: {Sentence}
Sentiment: {Label} | Positive / Negative |
| 2 | Input: {Sentence}
Prediction: {Label} | Positive / Negative |
| 3 | Review: {Sentence}
Sentiment: {Label} | good / bad |
| 4 | {Sentence} It was {Label} | good / bad |
| 5 | Review: {Sentence}
Positive Review: {Label} | Yes / No |
| 6 | Review: {Sentence}
Stars: {Label} | 5 / 0 |
| 7 | {Sentence} My overall feeling was that the movie was {Label} | good / bad |
| 8 | Review: {Sentence}
Question: Is the sentiment of the above review Positive or Negative?
Answer: {Label} | Positive / Negative |
| 9 | My review for last night's film: {Sentence} The critics agreed that this movie was {Label} | good / bad |

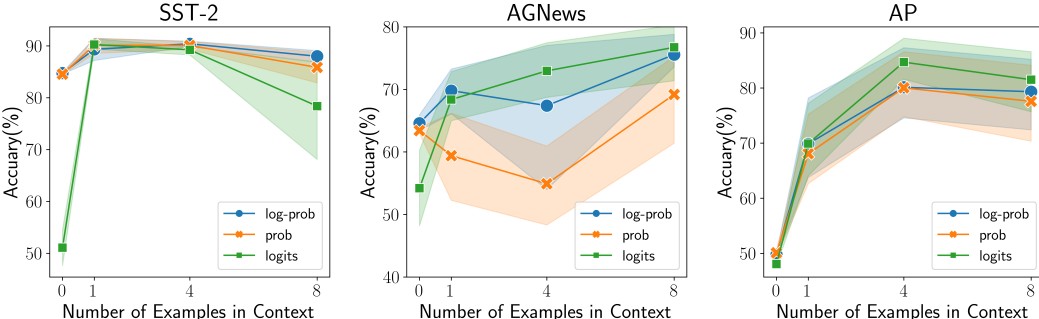

Figure 8: Different model output formats for estimation can influence the performance of PROCA, including logits, probability (prob), and log-probability (log-prob).

Table 9: Average performance of the conventional approach and PROCA for RoBERTa-Large across nine text classification tasks.

| Shot | Method | SST-2 | SST-5 | MR | Subj | AP | AGNews | DBpedia | RTE | TREC |
|------|--------|-------|-------|-----|------|-----|--------|---------|-----|------|
| 0-shot | RoBERTa-Large | $73.9_{0.0}$ | $30.7_{0.0}$ | $73.3_{0.0}$ | $60.9_{0.0}$ | $79.4_{0.0}$ | $65.7_{0.0}$ | $43.6_{0.0}$ | $51.3_{0.0}$ | $37.2_{0.0}$ |
| | PROCA | $\mathbf{90.4}_{0.2}$ | $\mathbf{45.4}_{1.9}$ | $\mathbf{85.8}_{0.2}$ | $\mathbf{68.8}_{0.2}$ | $\mathbf{79.7}_{1.4}$ | $\mathbf{69.0}_{0.8}$ | $\mathbf{79.9}_{0.8}$ | $\mathbf{58.7}_{3.5}$ | $\mathbf{42.8}_{2.2}$ |
| 8-shot | RoBERTa-Large | $\mathbf{94.1}_{0.4}$ | $40.8_{2.0}$ | $80.2_{7.7}$ | $51.9_{3.4}$ | $67.5_{5.8}$ | $63.7_{16.1}$ | $54.9_{17.6}$ | $57.8_{0.8}$ | $35.4_{2.7}$ |
| | PROCA | $93.8_{0.2}$ | $\mathbf{47.9}_{2.0}$ | $\mathbf{88.0}_{1.8}$ | $\mathbf{63.1}_{8.3}$ | $\mathbf{90.5}_{2.6}$ | $\mathbf{78.8}_{1.4}$ | $\mathbf{90.5}_{2.5}$ | $\mathbf{65.2}_{2.0}$ | $\mathbf{39.8}_{2.9}$ |

Table 10: Templates used in Appendix D.

| Task | Template | Label Space |
|------|----------|-------------|
| SST-2 |  Sentiment: [MASK] . | terrible / great |
| SST-5 |  Sentiment: [MASK] . | terrible / bad / okay / good / great |
| MR |  Sentiment: [MASK] . | terrible / great |
| Subj |  Type: [MASK] . | objective / subjective |
| TREC |  Type: [MASK] . | Number / Location / Person / Description / Entity / Abbreviation |
| AGNews |  Topic: [MASK] . | World / Sports / Business / Technology |
| DBPedia |  Topic: [MASK] . | Company / School / Artist / Athlete / Politician / Transportation / Building / Nature / Village / Animal / Plant / Album / Film / Book |

## C PROCA FOR BIDIRECTIONAL LANGUAGE MODELS.

To validate the effectiveness of PROCA for bidirectional language models, we conduct the evaluation on RoBERTa-Large (Liu et al., 2019) with 355M parameters across the nine text classification tasks. The templates for RoBERTa are constructed by replacing the label symbols in the templates in Table 6 with [MASK] symbols. We compare the 0- and 8-shot average performance of RoBerta-Large and PROCA over 5 random seeds and the detailed results are shown in Table 9. We can see that PROCA greatly outperforms RoBERTa-Large on all tasks except for 8-shot on SST-2. Therefore, we suggest that PROCA is also effective for Bidirectional Language Models.

Table 11: Performance comparison among domain conditional PMI, noisy channel method and PROCA when using GPT-2-Large on text classification tasks.

| Shot | Method | SST-2 | SST-5 | MR | Subj | AGNews | DBpedia | TREC |
|------|--------|-------|-------|-----|------|--------|---------|------|
| 0-shot | $\text{PMI}_{DC}$ | $\mathbf{85.6}_{0.0}$ | $22.0_{0.0}$ | - | - | $64.1_{0.0}$ | - | $44.0_{0.0}$ |
| | Channel | $76.9_{0.0}$ | $29.8_{0.0}$ | $72.8_{0.0}$ | $64.2_{0.0}$ | $59.8_{0.0}$ | $58.1_{0.0}$ | $\mathbf{46.4}_{0.0}$ |
| | PROCA | $84.6_{0.1}$ | $\mathbf{40.7}_{2.9}$ | $\mathbf{81.1}_{0.2}$ | $\mathbf{72.3}_{0.1}$ | $\mathbf{69.7}_{0.7}$ | $\mathbf{68.7}_{1.9}$ | $42.8_{0.9}$ |
| 8-shot | Channel | $\mathbf{86.4}_{1.2}$ | $35.3_{2.5}$ | $80.5_{2.5}$ | $49.9_{12.3}$ | $71.5_{3.6}$ | $72.5_{2.2}$ | $30.8_{6.3}$ |
| | PROCA | $\mathbf{86.4}_{3.8}$ | $\mathbf{37.7}_{5.2}$ | $\mathbf{85.5}_{1.2}$ | $\mathbf{82.4}_{6.0}$ | $\mathbf{75.7}_{2.1}$ | $\mathbf{87.4}_{3.7}$ | $\mathbf{51.8}_{8.4}$ |

## D PERFORMANCE COMPARISON WITH DOMAIN CONDITIONAL PMI SCORING FUNCTION AND NOISY CHANNEL METHOD.

Domain Conditional PMI Scoring Function (Holtzman et al., 2021) and Noisy Channel models (Min et al., 2021) also have strong performance on language model prompting in zero- and few-shot text classification. To compare against these approaches, we conduct the 0-shot and 8-shot evaluation on GPT-2-Large for channel models[2] and PROCA. We use simpler prompt templates for fair comparison and the details are shown in Table 10. The 0-shot accuracies of $\text{PMI}_{DC}$ reported by Holtzman et al. (2021) are also presented for further comparison. As shown in Table 11, it is clear that PROCA outperforms channel models in most cases and has competitive performance with $\text{PMI}_{DC}$ on the 0-shot scenario. $\text{PMI}_{DC}$ also calibrates the output distribution based on the domain premises and different domain premises may lead to different calibration results. In our opinion, channel models transform the decision boundary from label space to input space and it is more robust than the conventional language model in-context learning. There may still exist the decision boundary deviation problem in channel models. In comparison, our method directly calibrates the decision boundary by estimating the prototypical clusters and therefore boosts higher performance.

---

[2]The results of noisy channel models are replicated based on the released code https://github.com/shmsw25/Channel-LM-Prompting.

