# OpenReview forum: "Prototypical Calibration for Few-shot Learning of Language Models"
_ICLR.cc/2023/Conference — ICLR 2023 poster_

### Official Review · Reviewer_iRwh · 2022-10-24

**Confidence:** 2
**Correctness:** 2
**Technical Novelty And Significance:** 2
**Empirical Novelty And Significance:** 2
**Recommendation:** 5

**Clarity, Quality, Novelty And Reproducibility:**

For me, this submission omitts some useful information, making it difficult to understand the novelty of this work.

**Strength And Weaknesses:**

Strength: The analysis that decision boundary is of critical importance to the performance and Instability of few-shot demonstrations is interesting and suggestive. The authors propose prototypical calibration to adaptively learn a decision boundary for
few-shot classification of language models. They present provide rich experiments to show the effectiveness of their proposed PROCA on a wide range of text classification tasks.


Weaknesses: The paper is somewhat vague for me and I am still confused about some notions or symbols. For example, μn and Σn should be placed with vector and matrix symbols. I am confused the relationship between μn and μn,l. Does the {μn,l} is the l-th element of μn or they are different variables? Since tuple k is a cluster-label assignment, does this mean that k represents a matrix of N * N?  The authors transforme the assignment score to a weighted bipartite matching problem between N clusters and N labels. It would be better if the authors can provide more ditails about the  bipartite matching problem. In my opinion, the key problem in this work is learn cluster-label assignment, which however omits useful details and also overall algorithm. Since we are dealing a N way few shot problem, whether the few labeled samples in N way few shot task are utilized for the learning of assignment score. Besides, the authors utilize a small-scale unlabeled in-domain dataset, named as estimate set (Desti), to estimate the parameters. Does this mean the estimate set is sampled from "N way" without labeling?  In that cases, is the PROCA still fair for a typical N way few shot setting? Maybe the authors can clarify the estimate set  from this view.




**Summary Of The Paper:**

This work presents the analysis that decision boundary is of critical importance to the performance of few-shot demonstrations and the traditional decision boundary leads to the fragility of prompting LMs. The authors propose prototypical calibration to adaptively learn a more robust decision boundary for few-shot learning of language models.

**Summary Of The Review:**

Although calibrating prototypes for few-shot learning sounds reasonable, I have difficulty in understanding the specific approach of this submission, especially 3.2 and 3.3.

---

> ### Author Response · Authors · 2022-11-15
> **Response to Reviewer iRwh**
>
> We thank the reviewer for the constructive feedback and address the confusion below.
>
>
>
> > Confusion about some notions or symbols.
>
>
>
> In Section 3, $\mu_n$ denotes the mean vector of the $n$-th cluster. $\mu_{n,l}$ is the $l$-th element of $\mu_n$ that indicates how much the $n$-th cluster belongs to label $l$. We have also replaced $\mu_n$ and $\Sigma_n$ with vector and matrix symbols in the revised paper to make it more clear. Besides, the cluster-label assignment tuple $k$ is a permutation of \{1, 2, ..., N\} (actually an $N$-dimensional array rather than an $N\*N$ matrix) and $k_n$ represents the specific label that the $n$-th cluster is assigned to. For example, $k$=(2, 0, 1) means that cluster 0 is assigned to label 2, cluster 1 is assigned to label 0, and cluster 2 is assigned to label 1. So there are $N!$ possible assignments in total and we use Eq(5) to determine the optimal assignment $k^*$ for each estimation.
>
>
>
> > Details of the cluster-label assignment algorithm.
>
>
>
> The cluster-label assignment is only determined by the mean vectors  $\\{\mu_n\\}^N_{n=1}$ estimated on the unlabeled examples. None of the few-shot examples are used for assignment thus it is totally unsupervised, i.e., we cannot explicitly know the correspondence between $n$ clusters and $n$ labels. Fortunately, the N-th element of the mean vector of each cluster indicates how much it belongs to label $N$, which motivates us to solve the assignment problem through the weighted bipartite matching, where $\mu_{n,k_n}$ is the weight of the matching between cluster $n$ and label $k_n$. The optimal assignment is obtained by maximizing the assignment score according to Eq(5). There are $N!$ possible assignments so we adopt Kuhn-Munkres algorithm to accelerate it.
>
>
>
> > Does this mean the estimate set is sampled from "N way" without labeling?
>
>
>
> No. Directly sampling the estimate set from N-way means that we know the prior distribution of all labels in advance, and the data is balanced across N-way. We only simply sample small-scale unlabeled data from the training set, not according to N-way, so it is still a fair N-way few-shot setting. We also conduct experiments under different label proportions and Figure 5 shows that our method consistently surpasses the baseline under different unbalanced estimate sets. Furthermore, we explore another method that utilizes the generation ability of LMs to generate unlabeled estimate set. Table 2 shows that ProCa with generated estimate set also greatly outperforms the baseline.

---

> ### Author Response · Authors · 2022-12-10
> **Looking forward to post-rebuttal discussions**
>
> Dear reviewer iRwh,
>
> Thanks again for your constructive comments for helping us improve our paper in many aspects. We have provided detailed responses to address your concerns. The discussion deadline (Dec 12) is approaching. Your further feedback will be invaluable in improving our manuscript and we appreciate your time and expertise.
>
> Many thanks, The authors

---

### Official Review · Reviewer_f9YD · 2022-10-25

**Confidence:** 3
**Clarity, Quality, Novelty And Reproducibility:** The presentation is clear and quality…
**Correctness:** 4
**Technical Novelty And Significance:** 3
**Empirical Novelty And Significance:** 4
**Recommendation:** 8

**Strength And Weaknesses:**

Overall I believe this to be a very strong empirical investigation into in-context learning. The proposed approach is clearly motivated, simple and seems to be highly effective.

Strengths:
* Dramatic performance benefits from the proposed approach. 10+ pt improvement in most cases on average, 5+ in a few others (Table 1). ]
* Detailed empirical analysis that includes: effectiveness across templates, demonstration perturbation, class imbalance, estimate set construction, estimate set size,
* The results and analysis are clearly presented as are the methodological approach.
* Provides basis for future work on using using prototypes/clusters in the midst of in-context learning

Weaknesses:
* It might be interesting to consider how the fit of the GMM distribution effects the downstream task performance. Also since the number of points in the estimate set is not so many, it could be interesting to consider a wider array of clustering inference methods including more exhaustive/expensive methods.

**Summary Of The Paper:**

This paper presents a highly empirically effective approach for in-context learning using so-called _prototype calibration_. The simple, yet dramatically effective approach is well motivated by the authors and demonstrated to be empirically effective under a wide variety of settings, testing sensitivity to number of shots, prompt template, class imbalance and more.

**Summary Of The Review:**

This is a strong paper which presents a simple and effective approach for in context learning. The approach uses a GMM cluster-based approach to make predictions. The effectiveness of the approach is demonstrated through extensive experiments that evaluate the performance along many dimensions of number of shots, prompts, estimate set, etc.

---

> ### Author Response · Authors · 2022-11-15
> **Response to Reviewer f9YD**
>
> We thank the reviewer for acknowledging the contributions of our work.
>
>
>
> > How the fit of the GMM distribution affects the downstream task performance.
>
>
>
> We empirically observe that each prototypical cluster follows a Gaussian distribution.  Fitting the output distribution well is the prerequisite for calibrating the decision boundary into a more robust region and we find that GMM can do it well. We also propose the assignment score to select the optimal estimation which further boosts performance improvement.
>
> > Consider a wider array of clustering inference methods
>
>
>
> Based on empirical observations, we choose GMM clustering method which is simple and fast.
>
> As the reviewer suggests, the small scale of the estimate set gives us the chance to explore more exhaustive and expensive clustering methods. We believe that ProCa with these methods can achieve better estimation and boost performance further.

---

### Official Review · Reviewer_9WSu · 2022-10-28

**Confidence:** 5
**Correctness:** 4
**Technical Novelty And Significance:** 3
**Empirical Novelty And Significance:** 3
**Recommendation:** 6

**Clarity, Quality, Novelty And Reproducibility:**

The paper is written very clearly and easy to follow. The idea is novel and I believe it's easy to reproduce the results.

**Strength And Weaknesses:**

Strength
- Very strong and consistent empirical results from calibration. The performance is exceptionally strong for relatively smaller-sized models with fewer in-context learning examples.


Weaknesses
- There exist related works, though they do not necessarily claim to develop calibration approaches, they share the same spirit as this method to modify the output distributions for zero-shot or few-shot approaches [1][2]. Discussing these previous works and comparing results with them would help better position this paper.
- Related to the previous point, I think the related work section does not present the best relevant paper, especially the first section. I suggest the authors conduct a more thorough literature review, along with the direction of calibrating language models.
- I would also love to see if the calibration method works for masked language models given that MLMs have much stronger zero-shot performance on these evaluation datasets.

[1] Surface Form Competition: Why the Highest Probability Answer Isn’t Always Right
[2] Noisy Channel Language Model Prompting for Few-Shot Text Classification


**Summary Of The Paper:**

The paper proposes prototypical calibration for zero-shot and few-shot classification tasks when evaluating large language models. The approach first uses Gaussian mixture distribution to estimate the prototypical clusters for all categories of the classification task and then assigns each cluster to the corresponding label by solving a weighted bipartite matching problem. Each example’s prediction is calibrated by the likelihood of the prototypical clusters. They demonstrate significant improvements over contextual calibration (Zhao et al., 2021) on a wide range of tasks.


**Summary Of The Review:**

The paper has a novel idea for calibrating language models and shows strong empirical improvements over baselines. One concern is that the paper does not discuss a few critical related works or show empirical comparisons to these works.

---

> ### Author Response · Authors · 2022-11-15
> **Response to Reviewer 9WSu**
>
> We thank the reviewer for acknowledging the contributions of our work.
>
>
>
> > Discussion about critical related works.
>
>
>
> According to your suggestions, we did a more thorough literature review and added more discussions about the related works (including the two critical papers [1][2] you mentioned) in Section 5.
>
> The two recent papers [1][2] are closely related to ours. [1] explores the surface form competition problem in zero-shot models and proposes a new scoring function termed Domain Conditional Pointwise Mutual Information to reweight the answer scores. [2] introduces a noisy channel approach for language model prompting in zero- and few-shot text classification, which boosts significant improvement with lower variance. To the best of our knowledge, we are the first to study the instability of in-context learning from the perspective of decision boundary and our prototypical calibration outperforms prior direct calibration approach contextual calibration [3] by a large margin.
>
> Thank you so much for your constructive review to help us better position the paper.
>
>
>
> > If the calibration method works for masked language models.
>
>
>
> We conduct the evaluation on RoBERTa-Large(355M) across the nine text classification tasks to explore the effectiveness of our method for masked language models. The templates used for RoBERTa are constructed by replacing the label symbols in the templates in Table 6 with [MASK] symbols. We compare the 0- and 8-shot average performance of RoBerta-Large and our method over 5 random seeds. The detailed results are shown below. We can see that prototypical calibration greatly outperforms RoBERTa-Large in almost all cases. Therefore, we suggest that prototypical calibration is also effective for masked language models. The experimental results are also added to Appendix C of the revised version.
>
>
>
> | Shot   | Method        | SST-2 | SST-5 | MR | Subj | AP | AGNews | DBpedia | RTE | TREC |
> |--------|---------------|-------|-------|----|------|----|--------|---------|-----|------|
> | 0-shot | RoBERTa-Large | $73.9_{0.0}$ | $30.7_{0.0}$ | $73.3_{0.0}$ | $60.9_{0.0}$ | $79.4_{0.0}$ | $65.7_{0.0}$ | $43.6_{0.0}$ | $51.3_{0.0}$ | $37.2_{0.0}$       |
> | 0-shot | ProCa         | $\pmb{90.4}_{0.2}$ | $\pmb{45.4}_{1.9}$ | $\pmb{85.8}_{0.2}$ | $\pmb{68.8}_{0.2}$ | $\pmb{79.7}_{1.4}$ | $\pmb{69.0}_{0.8}$ | $\pmb{79.9}_{0.8}$ |  $\pmb{58.7}_{3.5}$ | $\pmb{42.8}_{2.2}$    |
> | 8-shot | RoBERTa-Large | $\pmb{94.1}_{0.4}$ | $40.8_{2.0}$ | $80.2_{7.7}$ | $51.9_{3.4}$ | $67.5_{5.8}$ | $63.7_{16.1}$ | $54.9_{17.6}$ | $57.8_{0.8}$ | $35.4_{2.7}$      |
> | 8-shot | ProCa         |  $93.8_{0.2}$ | $\pmb{47.9}_{2.0}$ | $\pmb{88.0}_{1.8}$ | $\pmb{63.1}_{8.3}$ | $\pmb{90.5}_{2.6}$ | $\pmb{78.8}_{1.4}$ | $\pmb{90.5}_{2.5}$ | $\pmb{65.2}_{2.0}$ | $\pmb{39.8}_{2.9}$    |
>
>
>
>
>
>
> [1] Holtzman, Ari, et al. "Surface Form Competition: Why the Highest Probability Answer Isn’t Always Right." EMNLP 2021.
>
>
>
> [2] Min, Sewon, et al. "Noisy Channel Language Model Prompting for Few-Shot Text Classification." ACL 2022.
>
>
>
> [3] Zhao, Zihao, et al. "Calibrate before use: Improving few-shot performance of language models." International Conference on Machine Learning. PMLR, 2021.

---

> > ### Comment · Reviewer_9WSu · 2022-11-16
> > **Thanks for the response!**
> >
> > Thanks the authors for providing empirical evidence in the effectiveness of the approach on MLMs! Could you explain why the zero-shot prompting results do not align with the results in the LM-BFF paper (https://arxiv.org/pdf/2012.15723.pdf)? I speculate that it might be because of prompts, but it would be nice for the authors to verify if it is the case.
> >
> > With regard to the other point, I understand that it could be an overarching requirement to compare to [1][2], but I do think having a deeper understanding about why ProCa outperforms/is comparable to [1][2] empirically would make the paper stronger.

---

> > > ### Author Response · Authors · 2022-11-17
> > > **Response to Reviewer 9WSu [1/2]**
> > >
> > > Thanks for your further response!
> > >
> > > > Verify the results on masked language models.
> > >
> > > According to your suggestions, we further evaluate our methods based on the prompt templates provided in [3] and the experimental results are below. We can see that both 0-shot results and 8-shot results align with those reported in [3] and our method also consistently outperforms the baseline. On the other hand, it suggests that our method is stable and insensitive to prompt templates.
> > > | Shot   | Method        | SST-2 | SST-5 | MR | Subj |RTE| TREC |
> > > |--------|---------------|-------|-------|----|------|----|-----|
> > > | 0-shot | RoBERTa-Large | $82.8_{0.0}$ | $35.0_{0.0}$ | $80.9_{0.0}$ | $49.6_{0.0}$ | $51.3_{0.0}$ | $\pmb{34.4}_{0.0}$ |
> > > | 0-shot | ProCa         | $\pmb{86.9}_{0.6}$ | $\pmb{42.7}_{1.6}$ | $\pmb{82.5}_{0.2}$ | $\pmb{66.6}_{0.6}$ |  $\pmb{58.7}_{3.5}$ | $32.3_{2.4}$ |
> > > | 8-shot | RoBERTa-Large | $84.9_{7.7}$ | $37.0_{7.2}$ | $78.1_{11.4}$ | $56.3_{10.5}$ | $57.8_{0.8}$ |$36.5_{10.9}$ |
> > > | 8-shot | ProCa         |  $\pmb{92.4}_{1.1}$ | $\pmb{38.7}_{6.5}$ | $\pmb{87.1}_{1.6}$ | $\pmb{71.8}_{0.8}$ |$\pmb{65.2}_{2.0}$ | $\pmb{44.8}_{7.6}$ |

---

> > > ### Author Response · Authors · 2022-11-17
> > > **Response to Reviewer 9WSu [2/2]**
> > >
> > > > Empirical comparison among $PMI_{DC}$ [1], Noisy Channel models [2] and Proca.
> > >
> > > We conduct the 0-shot and 8-shot evaluation on GPT-2-Large for channel models and ProCa. We use simpler prompt templates (as shown below) for fair comparison because ProCa calibrates the output distribution over label space and those prompt templates with task description words behind the label symbol are not suitable for ProCa. We find that the results of $PMI_{DC}$ based on these simple templates are not aligned with those reported in [1], so we present the reported 0-shot accuracies of $PMI_{DC}$ for comparison. As shown in the table below, we can see that ProCa outperforms channel models in most cases and has competitive performance with $PMI_{DC}$ on the 0-shot scenario. $PMI_{DC}$ also calibrates the output distribution based on the domain premises and different domain premises may lead to different calibration results. In our opinion, channel models transform the decision boundary from label space to input space and it is more robust than the conventional language model in-context learning. There may still exist the decision boundary deviation problem in channel models. In comparison, our method directly calibrates the decision boundary by estimating the prototypical clusters and therefore boosts higher performance.
> > >
> > > | **Shot** | **Method** | **SST-2** | **SST-5** | **MR** | **Subj** | **AGNews** | **DBPedia** | **TREC** |
> > > |:--------:|:----------:|:---------:|:---------:|:------:|:--------:|:----------:|:-----------:|:--------:|
> > > | 0-shot   | $PMI_{DC}$        |  $\pmb{85.6}_{0.0}$         | $22.0_{0.0} $     | -      | -        | $64.1_{0.0}  $     | -           | $44.0_{0.0}  $   |
> > > | 0-shot   | Channel    | $76.9_{0.0}$      | $29.8_{0.0} $     | $72.8_{0.0}$   | $64.2_{0.0}$     | $59.8_{0.0} $      | $58.1_{0.0} $       | $\pmb{46.4}_{0.0}$     |
> > > | 0-shot   | ProCa      | $84.6_{0.1}$      | $\pmb{40.7}_{2.9}$      | $\pmb{81.1}_{0.2}$   | $\pmb{72.3}_{0.1}$     | $\pmb{69.7}_{0.7}$       | $\pmb{68.7}_{1.9}$       | $42.8_{0.9}  $   |
> > > | 8-shot   | Channel    |$\pmb{86.4}_{1.2}$     | $35.3_{2.5} $     | $80.5_{2.5}$   | $49.9_{12.3}$     | $71.5_{3.6} $      |$72.5_{2.2}$      | $30.8_{6.3}$     |
> > > | 8-shot   | ProCa      | $\pmb{86.4}_{3.8}$     | $\pmb{37.7}_{5.2}$      | $\pmb{85.5}_{1.2}$   | $\pmb{82.4}_{6.0}$     | $\pmb{75.7}_{2.1}$      | $\pmb{87.4}_{3.7}$       | $\pmb{51.8}_{8.4}$     |
> > >
> > >
> > >
> > > | **Task** | **Template**            | **Label Space**                                                                                                                        |
> > > |----------|-----------------------------------|----------------------------------------------------------------------------------------------------------------------------------------|
> > > | SST-2    | \<S> Sentiment: [MASK] . | terrible / great                                                                                                                       |
> > > | SST-5    | \<S> Sentiment: [MASK] . | terrible / bad / okay / good / great                                                                                                   |
> > > | MR       | \<S> Sentiment: [MASK] . | terrible / great                                                                                                                       |
> > > | Subj     | \<S> Type: [MASK] .      | objective / subjective                                                                                                                 |
> > > | TREC     | \<S> Type: [MASK] .      | Number / Location / Person / Description / Entity / Abbreviation                                                                       |
> > > | AGNews   | \<S> Topic: [MASK] .     | World / Sports / Business / Technology                                                                                                 |
> > > | DBPedia  | \<S> Topic: [MASK] .           | Company  /  School  /  Artist / Athlete / Politician /   Transportation /  Building / Nature / Village /Animal / Plant / Album / Film / Book |
> > >
> > >
> > > [1] Holtzman, Ari, et al. "Surface Form Competition: Why the Highest Probability Answer Isn’t Always Right." EMNLP 2021.
> > >
> > > [2] Min, Sewon, et al. "Noisy Channel Language Model Prompting for Few-Shot Text Classification." ACL 2022.
> > >
> > > [3] Gao, Tianyu, Adam Fisch, and Danqi Chen. "Making Pre-trained Language Models Better Few-shot Learners." ACL 2021.

---

> > > ### Author Response · Authors · 2022-12-10
> > > **Looking forward to post-rebuttal discussions**
> > >
> > > Dear reviewer 9WSu，
> > >
> > > We would greatly appreciate it if you could let us know if we have addressed your concerns in the revised manuscript. If your concerns have not been resolved, could you please let us know which concerns were not sufficiently addressed so that we have a chance to respond before the discussion deadline (Dec 12). Your feedback will be invaluable in improving our manuscript.
> > >
> > > Many thanks, The authors

---

### Official Review · Reviewer_uD7y · 2022-11-04

**Confidence:** 3
**Correctness:** 3
**Technical Novelty And Significance:** 3
**Empirical Novelty And Significance:** Not applicable
**Recommendation:** 6

**Clarity, Quality, Novelty And Reproducibility:**

The paper is written clearly and with a good writing quality. The experimental section is also extensive and well written.

In terms of the originality, it is on the incremental side since 1-the decision boundary problem in few shot is known in the literature, 2-all the methods are borrowed from the literature. However, the authors did a good job with the presentation of the decision boundary problem and the way the methods are used and context can be considered as novel.

The authors provide the code, I skimmed through the lines but I did not run their code. I would say that it was reproducible from my impression, but I cannot comment in a certain way.

**Strength And Weaknesses:**

Strengths:

* The experimental results and applicability are well supported by using multiple datasets and their ablation study. Substantial improvements are obtained in terms of the accuracies.

* The idea of focusing on the decision boundary is well motivated and demonstrated.

Weaknesses:

* The label assignment part and multiple initializations and runs in the proposed algorithm can be computationally heavy as the authors mentioned. However, this might not be concerning since it is performed at the output space after the training.

*  Why the authors chose 4 and 8 shot in their experiments? It looks limiting and not reasoned well. Is it a standard procedure for these datasets?

* Although the authors mentioned and used averaging and careful initializations, the stability of the method is still a bit concerning to me. Too much unstable/sensitive methods are combined and dataset and task itself also poses another sensitivity issue by nature and since they are scarce. However, the results suggest otherwise unless it is 0 or 1 shot.

* It would be good to explore the boundaries of the method in terms of the original dataset/labels and the desired dataset/task, and a notion of closeness between these tasks.


**Summary Of The Paper:**

The paper explores the importance of the decision boundary in few-shot learning in language models with GPT. Then, the authors propose an improvement over the existing few shot methods by adaptively learning this decision boundary. The decision boundary is learned by modeling the output distributions of GPT few-shot model as GMM and fitting to the class labels. The results shown are better than the baselines up to 13%. The ablation study is done by measuring the effect of template choices, class imbalance and class permutations. The robustness is measured by running with several random initializations.

**Summary Of The Review:**

The authors propose a method where they learn decision boundary for an in context GPT few shot learning method. The motivation, story and the experiments parts are the strongest side of the paper. The ideas and results are plenty and well supported, which surpass the disadvantage coming from the computational complexity.

---

> ### Author Response · Authors · 2022-11-15
> **Response to Reviewer uD7y**
>
> We thank the reviewer for acknowledging the contributions of our work and address the confusion below.
>
>
>
> > The label assignment and multiple initializations and runs can be computationally heavy.
>
>
>
> The computational cost of the label assignment is negligible since the class number is very small (2-14). Multiple repetitions also bring little additional cost because a small-scale estimate set is sufficient for ProCa. Specifically, the time cost per estimation on one CPU for 2-way tasks (like SST-2) is less than 0.2 seconds and that for DBpedia (14-way) is less than 3 seconds.
>
>
>
> > Why choose 4- and 8-shot in the experiments?
>
>
>
> We follow the prior work Contextual Calibration [1] to choose 0-, 1-, 4-, and 8-shot for the performance comparison and our method performs consistent improvements for other shots.
>
>
>
> >  The stability of the method.
>
>
>
> From the perspective of our method, three factors may affect the stability: the number of estimation repetitions, the balance and the size of the estimate set (the cluster-label assignment is deterministic for each estimation and thus it does not affect the results).
>
>
>
> 1. About the number of estimation repetitions. We found that the performance of ProCa is very stable after 100 estimation repetitions with different initializations. Especially for those two-way classification tasks, only 10 repetitions are enough. Since the computational cost is low, we use 100 repetitions for all tasks.
>
>
>
> 2. About the balance of the estimate set. As shown in Figure 5, our method outperforms ConCa by a large margin at all class imbalance ratios.
>
>
>
> 3. About the size of the estimate set. From Figure 7, we observe that a small-scale estimate set can guarantee the stable performance improvement of ProCa.
>
>
>
> From the perspective of in-context learning itself, the factors that affect the stability include demonstration choices, templates, the permutation and balance of demonstrations.
>
>
>
> 1. About demonstration choices. To conduct fair and robust comparisons, all results in our paper are conducted under five different random seeds. From the experimental results in Table 1, ProCa performs higher average performance under different demonstration choices with lower standard deviation, which indicates that our method can improve the stability of in-context learning.
>
>
>
> 2. About different templates. Figure 4 shows that ProCa can improve both the performance and the stability across various templates compared with ICL baseline and ConCa.
>
>
>
> 3. About the permutation and balance of demonstrations. We conduct experiments under different label proportions and permutations of demonstrations. Figure 6 shows the stability and effectiveness of ProCa.
>
>
>
> Overall, although there are some factors that can possibly affect the results especially under 0- and 1-shot settings, through a series of ablations and comprehensive comparisons, we demonstrate ProCa achieves better stability while improving performance.
>
>
>
> [1] Zhao, Zihao, et al. "Calibrate before use: Improving few-shot performance of language models." International Conference on Machine Learning. PMLR, 2021.

---

### Decision · Program_Chairs · 2023-01-20

**Decision:**

Accept: poster

**Justification For Why Not Higher Score:**

+ Novelty is somewhat incremental
+ Related works are not explained much or compared against in experiments
+ Method can be potentially unstable under several settings

**Justification For Why Not Lower Score:**

The reviewers found the work generally well written, and the empirical results from calibration are strong and quite convincing.

**Metareview: Summary, Strengths And Weaknesses:**

This work proposes an alternative to naive greedy decoding during generation in few-shot learning. Namely, instead of simply selecting the token with highest probability, they first adopt a Gaussian mixture model to estimate the distribution for different categories under clusters, then perform weighted bipartite matching to match clusters to corresponding labels so as to calibrate the decision boundaries.

The reviewers found the work generally well written, and the empirical results from calibration are strong and quite convincing.

All reviewers leaned toward accept with the exception of one reviewer who gave a score with low confidence and whose primary criticism was lack of clarity. Unfortunately, they also did not respond during the discussions. I agree with the majority consensus.

**Note From Pc:**

if the above contains the word "oral" or "spotlight" please see: "oral" presentation means -> notable-top-5% and "spotlight" means -> notable-top-25%. As stated in our emails, we are disassociating presentation type from AC recommendations